# Long-Term Effects of Vitamin D Supplementation in Obese Children During Integrated Weight–Loss Programme—A Double Blind Randomized Placebo–Controlled Trial

**DOI:** 10.3390/nu12041093

**Published:** 2020-04-15

**Authors:** Michał Brzeziński, Agnieszka Jankowska, Magdalena Słomińska-Frączek, Paulina Metelska, Piotr Wiśniewski, Piotr Socha, Agnieszka Szlagatys-Sidorkiewicz

**Affiliations:** 1Department of Paediatrics, Gastroenterology, Allergology and Paediatric Nutrition, Medical University of Gdańsk, 80–803 Gdańsk, Poland; ajankowska@gumed.edu.pl (A.J.); agnieszka.szlagatys-sidorkiewicz@gumed.edu.pl (A.S.-S.); 2Department of Public Health and Social Medicine, Medical University of Gdansk, 80–210 Gdańsk, Poland; metelska.paulina@gumed.edu.pl; 3Department of Paediatrics, Copernicus Medical Center, 80-462 Gdańsk, Poland; madds@wp.pl; 4“6–10–14 for Health”–University Clinical Center, 80-952 Gdańsk, Poland; 5Department of Endocrinology and Internal Medicine, Medical University of Gdansk, 80–952 Gdańsk, Poland; piotr.wisniewski@gumed.edu.pl; 6Department of Gastroenterology, Hepatology and Feeding Disorders, The Children’s Memorial Health Institute, 04–730 Warszawa, Poland; p.socha@ipczd.pl

**Keywords:** vitamin D, obesity, weight–loss, body composition

## Abstract

Background: Vitamin D was studied in regards to its possible impact on body mass reduction and metabolic changes in adults and children with obesity yet there were no studies assessing the impact of vitamin D supplementation during a weight management program in children and adolescence. The aim of our study was to assess the influence of 26 weeks of vitamin D supplementation in overweight and obese children undergoing an integrated 12–months’ long weight loss program on body mass reduction, body composition and bone mineral density. Methods: A double–blind randomized placebo–controlled trial. Vitamin D deficient patients (<30 ng/ml level of vitamin D) aged 6–14, participating in multidisciplinary weight management program were randomly allocated to receiving vitamin D (1200 IU) or placebo for the first 26 weeks of the intervention. Results: Out of the 152 qualified patients, 109 (72%) completed a full cycle of four visits scheduled in the program. There were no difference in the level of BMI (body mass index) change – both raw BMI and BMI centiles. Although the reduction of BMI centiles was greater in the vitamin D vs. placebo group (−4.28 ± 8.43 vs. −2.53 ± 6.10) the difference was not statistically significant (*p* = 0.319). Similarly the reduction in fat mass—assessed both using bioimpedance and DEXa was achieved, yet the differences between the groups were not statistically significant. Conclusions: Our study ads substantial results to support the thesis on no effect of vitamin D supplementation on body weight reduction in children and adolescents with vitamin D insufficiency undergoing a weight management program.

## 1. Introduction

Worldwide prevalence of overweight and obesity is increasing both in adults and children [1,2]. It is observed on all continents, with the highest burden in those from lower socioeconomic and educational groups [1,3]. Obesity among others is associated with a wide range of metabolic disorders, such as insulin resistance, hyperinsulinemia, and impaired tolerance of glucose, abnormal fasting plasma glucose, symptomatic diabetes mellitus, lipid disorders and cardiovascular disorders, namely arterial hypertension [4,5,6]. Obesity alone is responsible for a significant increase in the risk of mortality in general population with [7]. In view of high risk of complications resulting from childhood obesity, early implementation of intervention programs seems to be vitally important, as in children and adolescents it is the first–choice intervention, although with limited effectiveness [8]. Several previous studies showed that integrated multidisciplinary weight–loss programs, which include the child’s family as well, are the most effective [9,10,11]. Reduction of fat mass is associated with normalization of metabolic parameters, such as inflammatory markers, lipid profile, insulin resistance and arterial blood pressure [12,13,14]. Therefore, early and efficient intervention increases likelihood of staying healthy in the future. As pharmacological and surgical interventions in children are limited [15,16,17], trials looking for substances supporting lifestyle interventions were run, looking at several different dietary supplements, hers (green tea, yerba mate), DHA (docosahexaenoic acid) among others [18,19,20,21,22]. Vitamin D was also studied in regards to its possible impact on body mass reduction and metabolic changes in adults and children with obesity [23,24].

The role of vitamin D in energetic metabolism has been emphasized recently. Obese children frequently present with low blood concentrations of vitamin D [25,26,27]. This probably results from lower dietary intake of this vitamin by obese individuals on the one hand, and less outdoor physical activity on the other [28,29,30]. Moreover, a higher percentage of fat mass is associated with a lower blood concentration of vitamin D, which may, at least partially, result from its sequestration in adipose tissue [31]. Animal experiments with labelled vitamin D showed that this vitamin is accumulated in adipose tissue and slowly released to blood [32]. 

Consequently, a vitamin D deficiency in obese children seems to be associated with a significant increase of risk of many metabolic disorders associated with obesity, such as insulin resistance, hyperinsulinemia, impaired tolerance of glucose, abnormal fasting plasma glucose, symptomatic diabetes mellitus, lipid disorders and cardiovascular morbidity, namely arterial hypertension [5,6]. There is a number of observational studies which demonstrate the substantial role of vitamin D deficiency in developing metabolic syndrome and other complications of obesity [33,34,35]. However, we lack interventional studies to link these observations to demonstrate a causal relationship.

In this study we wanted to assess the influence of 26 weeks of vitamin D supplementation in overweight and obese children undergoing an integrated 12–months’ long weight loss program on body mass reduction, body composition and bone mineral density [36].

## 2. Materials and Methods 

Detailed information about the study protocol, participants and design of the interventional program has been previously published [36]. All study participants were recruited from “6–10–14 for Health” program run by University Clinical Center in Gdansk, Poland. The program is a multidisciplinary, interventional program dedicated to children aged 6–15 and their parents. All school children in Gdansk aged 6–15 are screened in a 3–4–year–interval by dedicated teams (pediatrician and or nurses). All children with BMI centile above 85^th^ centile are invited to the “6–10–14 for Health” program. All participants of the “6–10–14 for Health” program and their family members (parents/caregivers) were offered 12–month integrated intervention, including individual medical (pediatric), dietetic, physical activity and psychological counselling, during one meeting (4 x 20–25min). This intervention was offered to all the participants in a 0–3–6–12 month scheme.

### 2.1. Trial Design

A double–blind placebo–controlled trial. Vitamin D deficient patients (<30 ng/ml level of vitamin D) participating in multidisciplinary weight program were randomized to two arms (1:1 ratio): receiving vitamin D (1200 IU) or placebo for the first 26 weeks of the intervention. 

We hypothesized that the supplementation with vitamin D in obese children showing low serum 25(OH)D3 during weight–loss program could positively influence body mass index (BMI), muscle mass, bone mass and mineral density and biochemical markers of metabolic complications related to obesity compared to placebo.

The study was conducted at the Department of Paediatrics, Paediatric Gastroenterology, Allergology and Children Nutrition, the Medical University of Gdansk and within the framework of “6–10–14 for Health” Interventional Programme, University Clinical Centre in Gdansk. 

### 2.2. Participants

Eligible participants were children aged 6, 9–11 and 14, according to the “6–10–14 for Health” program protocol:

**Inclusion criteria:** overweight (BMI ≥ 85th < 95th percentile) or obesity (BMI ≥ 95th percentile), identified on the basis of anthropometric parameters using Polish reference centile charts—OLAF project [37]; blood concentration of 25(OH)D3 < 30 ng/ml; written informed consent of legal caregivers.

**Exclusion criteria:** Chronic conditions (asthma or allergies, inflammatory diseases of connective tissue, gastrointestinal disorders, diseases of kidneys and liver, disorders of bone metabolism); contraindications to administration of vitamin D; administration of any preparation containing vitamin D, calcium, or steroid hormones during three months preceding the study.

### 2.3. Ethics Approval and Consent to Participate

The study is approved by Independent Bioethics Committee for Scientific Research at Medical University of Gdansk, Poland, [NKBBN/130–206/2015] dated 25.05.2015. The parents/caregivers gave their written consent before the start of any study procedure. The study protocol was peer reviewed by financial supporter (Fundacja Nutricia) independent commission, during the grant application process. Trial registration no: NCT 02828228; trial registration date: 8 June 2016 registered in: ClinicalTrials.gov.

### 2.4. Study Procedure

The study period covered four appointments within the “6–10–14 for Health” obesity management program at the 0, 3, 6 and 12 month mark. All visits included individual meetings with a pediatrician, dietician, physical activity specialist and psychologist. All child/parent dyads had individual meetings with all 4 specialists—one directly after other. Detailed information about the program can be found in previous publications [36]. During the first appointment, the caregivers of all children were asked to provide written consent to the child taking part in the trial. Refusal to participate in the trial did not influence participation in the interventional obesity management program. Children with low blood concentration of 25(OH)D3 (<30 ng/ml) were referred for DXA (dual–energy x–ray absorptiometry) within two weeks from the starting visit.

The enrolled subjects were randomly assigned to one of the two groups using computer generated randomization tables. Then participants were randomly assigned to one of the two trial groups:GROUP I (Vitamin D group): medical intervention, intervention of dietician, psychologist and physical education specialist, parental education + oral administration of vitamin D3 (1200 i.u. daily) for 26 weeksGROUP II (Placebo group): medical intervention, intervention of dietician, psychologist and physical education specialist, parental education + daily oral administration of placebo for 26 weeks.

Study time line chart is shown in Figure 1.

### 2.5. Randomization and Blinding

The randomization list was generated by Office of Clinical and Scientific Research, University Clinical Centre (OCSR UCK), with no clinical involvement in the trial, via a computer program (StatsDirect) with an allocation ratio of 1:1 and with a block of 6. The allocation sequence was concealed from the researchers responsible for enrolling and assessing participants. Throughout the duration of the study, all investigators, participants, outcome assessors, and data analysts were blinded to the assigned treatment. Allocation to groups and drug/placebo distribution was performed by an independent researcher (M.S–F) not directly involved in the interventional program.

### 2.6. Treatment Dispensing and Assessment of Compliance

Both of the study treatments: vitamin D (1200 IU) and placebo were provided by the company (Sequoia) in identical capsules and packages (5 capsules in one blister, 6 blisters in a box). The sets of 7 boxes (6 months treatment) were prepared and blinded by Office of Clinical and Scientific Research, University Clinical Centre. Study treatment was dispensed by the investigator at the enrolment visit. At the final visit the sets of blisters and boxes were retrieved from the patient. The number of remaining capsules were documented to assess the compliance.

### 2.7. Outcome Measures

Primary endpoint: Change in BMI, BMI centile after 26 weeks of vitamin D supplementation. Secondary endpoints: change in body composition and bone mineral density and vitamin D level after 26 weeks of vitamin D supplementation. Additionally, we assessed changes in blood level of vitamin D.

### 2.8. Sample Size

Assuming probability of the event (at least 10% reduction in baseline BMI over the follow–up period) at 0.85 and 0.6 for the experimental and control group, respectively, minimum sample size providing 0.9 statistical power for alpha equal 0.05 or lower and beta equal 0.1 or lower was estimated at 130 (65 per group).

### 2.9. Statistical Analysis

Normal distribution of continuous variables was verified with the Shapiro–Wilk test. Descriptive statistics are presented as the mean or median and standard deviation from the mean. Between groups comparisons were carried out using the Mann–Whitney U test. Nonparametric tests were chosen because of the large number of significant Shapiro tests, which were used for normality assumption assessment. All statistical tests were two–tailed and performed at the 5% level of significance. All analyses were performed on the intention–to–treat basis, in which all of the participants in a trial are analyzed according to the intervention to which they were assigned, analyzing only participants who completed the whole weight management intervention program. Statistical analyses were performed with the Statistica 13 (TIBCO Software Inc., Tulsa, USA 2017).

## 3. Results

After obtaining and verifying the correctness of the recorded data (cross-checked with source data of 30% of patients), the data was decoded according to the protocol received from OCSR UCK in Gdansk.

After decoding, a full analysis of available anthropometric data and the results of densitometric examination was performed for patients divided into groups—receiving placebo and vitamin D.

A total of 170 patients were eligible for enrolment in the study during the enrolment period. A total of 152 patients were eventually included in the study (*n* = 8—no parental consent, *n* = 10—vitamin D intake during three months before the study). Subsequently the patients were randomly assigned to groups: 67 to the placebo group and 85 to the Vitamin D group. Out of the 152 qualified patients, 109 (72%) completed a full cycle of four visits scheduled in the program. In the placebo group, 64 patients completed the active phase of 26 weeks of supplementation, 53 completed the comprehensive treatment program (52 week duration).

The patient flow is shown in Figure 2.

Basic demographic and clinical data for both groups are presented in Table 1.

There were no statistically significant differences between the vitamin D and placebo groups at the start of the study.

The differences between the groups in terms of main endpoints were subsequently assessed.

The results of basic anthropometric and 25(OH)D concentration tests are shown in Table 2. 

The results of bioelectric impedance measurements are shown in Table 3. 

The results of the dual X–ray absorptiometry (DXA) measurements are presented in Table 4. 

Both groups had a reduction in BMI centiles. Although the reduction was greater in the vitamin D vs. placebo group (–4.28 ± 8.43 vs. –2.53 ±6.10) the difference was not statistically significant (*p* = 0.319). Similarly the reduction in fat mass—assessed both using bioimpedance and DEXa was achieved, yet the differences between the groups were not statistically significant, as shown in Table 3 and Table 4.

The analysis showed statistically significant differences between the groups only in 25(OH) D3 concentration in the measurements taken after the supplementation period (24.99 vs. 16.25; *p* = 0.000) and in the difference between second and first measurement of vitamin D levels (6.06 vs. –4.24; *p* = 0.000), and in the difference between second and first measurement of bone mineral density in the spine (Sp BMD) (0.04 vs. 0.06; *p* < 0.0256). The difference was higher in the placebo group. 

There was no difference between the placebo group and the vitamin D group in BMI reduction, BMI centile, fat tissue in kg, % of fat tissue (assessed both by performing BIA and densitometry analysis).

Additionally, we have performed an analysis of correlations (Spearman rank correlation coefficient) between vitamin D levels at Visit 1 and 4 and changes of vitamin D level between the visits dependent on several anthropometric variables. Results are presented in Table 5. 

The results show that there were no important correlation between the initial BMI centile and vitamin D level at the first visit. Additionally the correlation did not show any significant influence of BMI changes on changes in vitamin D level after the supplementation period. There were significant negative correlations (*p* < 0.05) between fat mass % and levels of vitamin D during both visits—which can confirm the relation between the fat mass and blood level of vitamin D. Yet this was a rather weak correlation (–0.25 to –0.21). Further, it was not confirmed in DXA measurements.

## 4. Discussion

### 4.1. Effect of Vitamin D Supplementation on Body Mass Reduction

Presented study is the first randomized trial to assess potential effects of vitamin D supplementation in body mass reduction in overweight and obese children. Results of present study show that supplementation of vitamin D did not have a statistically significant, put potentially clinically important, influence on body mass (BMI, BMI centile) body composition or bone mineral density comparing to placebo groups during an organized obesity management program in children. 

Biological role of vitamin D in etiopathogenesis of metabolic syndrome represents an interesting issue. Previous studies conducted among children revealed inverse relationship between blood concentration of vitamin D and waist circumference, systolic blood pressure, insulin resistance, fasting glucose, total cholesterol, triglycerides and LDL cholesterol, as well as positive association between the concentration of vitamin D and HDL cholesterol [25,38,39]. It seems that vitamin D can interfere with secretion of insulin both directly—binding to its receptors [VDR] on pancreatic β cells, and indirectly by modulating concentration of calcium in extracellular space [40].

Importantly, a positive association between the concentration of vitamin D and sensitivity to insulin was observed in obese children, along with an inverse relationship between the level of this vitamin and concentration of glycated hemoglobin (HbA1c) [41]. Supplementation with vitamin D in obese adolescents resulted in decrease of insulin resistance, while levels of inflammatory markers [CRP, TNF–α, IL–6] remained unchanged [42]. Till now, to the best of our knowledge, no studies were carried out in children or adolescents regarding the effects of vitamin D supplementation on body mass reduction during an organized lifestyle modification program. As we presented, adding a 1200 IU/day dose of vitamin D did not lead to higher changes in BMI (BMI centile) or fat mass as well as fat free mass changes in children aged 6–14. It needs to be stated that the results presented show that although there was a reduction in BMI in vitamin D group compared to placebo group (–0.46 ± 1.80 vs. 0.11 ±1.84) and BMI centiles also showed higher reduction in children supplemented with vitamin D (–4.28 ± 8.43 vs. –2.53 ±6.10) none of those results was statistically significant (*p* = 0.203 and *p* = 0.319 respectively). Similar results can be found in bioimpedance measurements, but not DXA assessment. Presented results would be presented as clinically valid. Although this 26-week long supplementation had an influence on blood concentration level of 25(OH) D in the active treatment group, only 6 out of all patients reached a level above 30 ng/ml in the final assessment (2 in the placebo group) 52 weeks after the start of the intervention. This shows that vitamin D can potentially have an impact on weight loss level but possibly due to resignation ratio or sample size we were not able to show that effect.

As previous studies in adolescents and adults showed, low (~1000 IU/daily) or high doses (up to 300,000 IU/month) of vitamin D supplementation have a very mixed results in influencing changes in fat mass, free fat mass or muscle mass [43,44,45]. Data regarding effect of vitamin D supplementation on body mass/body fat changes are limited to studies and meta–analyses/ reviews regarding adults with wide range of interventions (medical weight loss, bariatric surgery, low–caloric diet) were used together with vitamin D supplementation. Additionally those studies were focused on finding the optimal dose of supplementation to reach the optimal (>30 ng/l) level of vitamin D [44] or assessing the association of vitamin level and body fat. Meta-analysis of studies in obese adults showed that although there was an impact of vitamin D supplementation on body fat, the results were not statistically significant [43]. This study also acknowledges that the impact of vitamin D supplementation on body fat reduction has a linear effect up to 2000 IU/d, with no benefits in increasing the dose in adults. The only review regarding impact of vitamin D supplementation on bone mineral density was focused on general population—as no studies was directly focused on overweight/obese children. In addition, no sub–analysis was performed to show the effect depending on body mass. Winzenberg states that the impact of vitamin D supplementation can be higher in vitamin D deficient children, but the effect is small—in most studies standardized mean difference between groups was small (<0,3) [45]. Yet all of these measures are only proxy measures to changes in body weight as a primary outcome of supplementation of vitamin D in obese children/adolescents. As for now, no studies have demonstrated effectiveness of such a strategy for supporting body fat reduction throughout a long–term interventional process. 

### 4.2. Effect of Vitamin D Supplementation on Bone Mineral Density During Weight Loss

Metabolic effects of obesity on growth and maturation of bones are still not fully understood. Moreover, the results of previous studies analyzing bone mass and density in obese individuals are highly inconclusive. While some authors claimed a decrease in bone mass relative to body weight [46], others did not document significant differences in bone mineral density [47] or showed an increase in body mass and bone size in obese children, adolescents and adults. Increased bone mass and density observed in obese individuals is postulated to be a response to greater mechanical load, direct influence of leptin or enhanced enzymatic activity of aromatase [48,49]. Nevertheless, obesity markedly increases the risk of bone fractures in children [50]. Vitamin D plays important biological role in the process of bone maturation and mineralization. Previous studies documented an inverse relationship between blood concentration of vitamin D and bone mineral density [51,52]. One meta–analysis revealed that supplementation of vitamin D can improve both bone mineral density and bone mass in individuals with low blood levels of this vitamin [45]. The effects of supplementation are particularly favorable in premenarcheal girls with normal body weight, in whom administration of vitamin D resulted in increases of both bone mass and fat–free mass [53]. An analysis of 58 morbidly obese teenagers showed that individuals with physiological blood concentration of PTH (parathyroid hormone) have normal bone mineral density, irrespectively of their vitamin D levels [54]. In contrast, a recently published study involving a small group of adolescents with obesity (*n* = 24) and normal body weight (*n* = 25) showed that obese people present with higher bone mineral density, irrespectively their blood concentration of vitamin D and despite lower level of physical activity than their normal-weight peers. Moreover, the differences in bone mineral density turned out to be independent from fat-free mass content. Furthermore, bone mineral density was associated with blood concentrations of leptin and insulin [55]. Our study shows clearly that we can observe an increase of bone mineral density both spinal and subtotal BMD with body mass reduction. Yet there is no impact of vitamin D supplementation on the level of bone density assessed using DXA. This shows that the body mass reduction itself impacts the bone mineralization the most.

Apart from many unquestioned favorable health effects of losing excessive weight, this process may also be associated with enhanced bone turnover and decrease in bone mineral density. In recently published systemic review, the decrease of bone mass was reported following calorie-restricting diet but not in exercise–induced weight-loss [56]. However, this evidence originates mostly from studies conducted among adults [57,58,59], and to the best of our knowledge, the issue in question was a subject of only one study of adolescents after bariatric surgeries [60]. The results from previous intervention studies suggest that a low-calorie albeit high-protein (ca. 30%) diet, containing high amounts of dairy products, can prevent the loss of bone mass and a decrease in bone mineralization [61]. Yet supplementation with high doses of vitamin D can have a negative effect on bone mineralization in adults—as a recent study by Burt et al. showed [62].

Study limitations:

This study has some limitations that need to be taken into account when assessing the usefulness of the results:

– Children aged 6–14 years old were included in the study—this is not a homogenic group when it comes to maturation/puberty status—and this has an impact on both the ability to decrease body mass and bone turnover and mineralization;

– We were giving one dose of vitamin D (1200 IU) to all participants independent the body mass and age, which could have result in less effective increase of 25(OH)D level, which in turn could have impacted the changes in body mass or bone mineral density;

– 26 weeks may be too short to establish the effects of vitamin D supplementation on the rate of skeletal mineralization

– There was an almost 30% drop–out, seen especially in the second part of the program—after finishing the active supplementation with vitamin D/placebo period. This was unlikely to be due to the treatment itself (as there were no important side effects registered). The level of lost to follow-up patients in the interventional program is similar to other such programs seen in Poland and other European countries [63,64].

– Finally, the sample sizes were smaller than anticipated and there is a possibility that with the estimated reduction of 10% of BMI reduction the study sample is underestimated. It is possible that the study was underpowered to detect smaller changes in BMI/BMI centiles as well as other parameters.

## 5. Conclusions

Available data on the efficacy of vitamin supplementation during weight loss are inconclusive and mostly limited to adults. Our study shows that there is a limited or no effect of vitamin D supplementation on body weight reduction in children and adolescents with vitamin D insufficiency. Being aware and understanding the potential limitation of our study—wide age group, one dose of vitamin D, small sample of the study we believe that further research in this field is needed.

## Figures and Tables

**Figure 1 nutrients-12-01093-f001:**
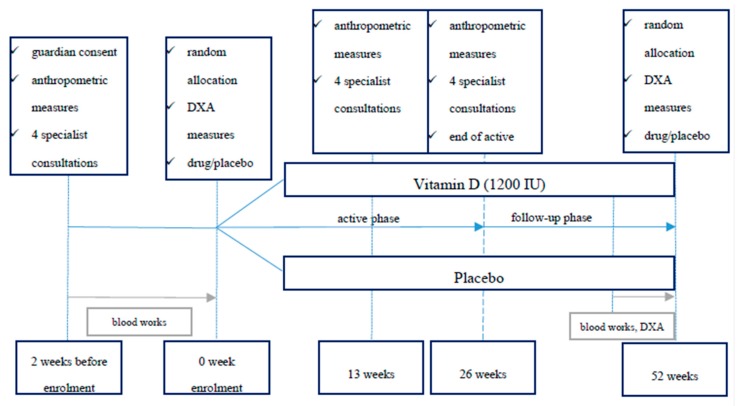
Patients’ time line in the study.

**Figure 2 nutrients-12-01093-f002:**
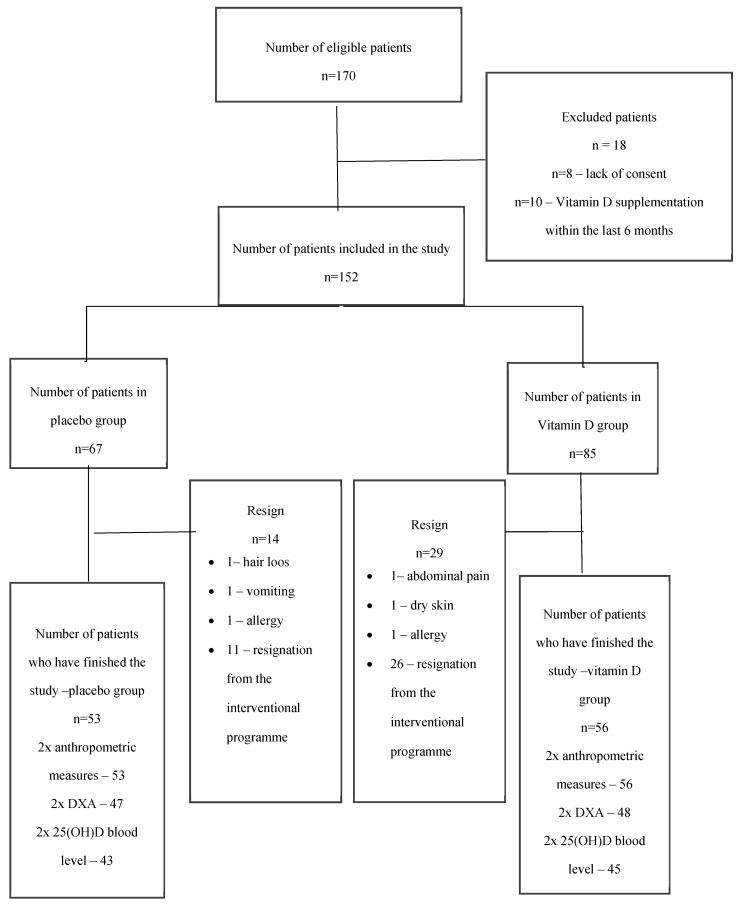
Patient flowchart in the study

**Table 1 nutrients-12-01093-t001:** Basic anthropometric and clinical data of studied groups (vitamin D and placebo) at visit 1.

	vitamin D (*n* = 85)		placebo (*n* = 67)		
	Mean ± SD	(95% CI)	Mean ± SD	CI (–95%)	*p*
age (years)	11.10 ± 2.84	10.49–11.72	10.70 ± 3.13	9.92–11.47	0.389
body mass (kg)	59.01 ± 21.04	54.47–63.55	56.89 ± 20.08	52.00–61.9	0.706
height (cm)	150.79 ± 16.69	147.19–154.39	149.31 ± 18.29	144.85–153.77	0.660
BMI	24.97 ± 4.12	24.08–25.86	24.53 ± 3.57	23.66–25.41	0.759
BMI centile	95.18 ± 3.24	94.49–95.88	95.23 ±3.43	94.41–96.06	0.812
no. of girls at visit 1	46		38		0.827
% of girls at visit 1	54.12%		56.72%		

*p* < 0.05 Mann–Whitney U test analysis.

**Table 2 nutrients-12-01093-t002:** Results of basic anthropometric and 25(OH)D concentration tests in studied groups.

Variable	vitamin D			placebo			
	No	Mean ± SD	95% CI	No	Mean ±	95% CI	*p*
BMI visit 1	85	24.97 ± 4.12	24.08–25.86	67	24.53 ± 3.58	23.66–25.41	0.759
BMI visit 4	56	24.33 ± 3.97	23.27–25.39	53	24.68 ± 3.46	23.73–25.64	0.479
ΔBMI visit 4–1	56	–0.46 ± 1.80	−0.94–0.03	53	0.11 ±1.84	−0.40–0.61	0.203
BMI centile 1 visit	85	95.18 ± 3.22	94.49–95.88	67	95.23 ± 3.39	94.41–96.06	0.812
BMI centile 4 visit	56	90.91 ± 9.40	88.39–93.43	53	92.64 ± 7.53	90.56–94.71	0.303
Δ in BMI centiles visit 4–1	56	−4.28 ± 8.43	−6.54–−2.03	53	−2.53 ± 6.10	−4.21–−0.85	0.319
25 (OH) D level visit 1	85	19.35 ± 5.46	18.16–20.55	66	19.79 ± 5.15	18.52–21.06	0.622 *
25 (OH) D level visit 4	45	24.99 ± 5.54	23.33–26.66	43	18.3 ± 6.70	16.25–20.37	0.000 *
Δ 25 (OH) level visit 4–1	45	6.06 ± 5.80	4.32–7.81	43	−2.40 ± 5.97	−4.24–−0.57	0.000 *

*p* < 0.05 Mann–Whitney U test, * t–student test for independent samples.

**Table 3 nutrients-12-01093-t003:** Results of bioelectric impedance measurements in studied groups.

Variable	vitamin D			placebo			
	No	Mean	CI (–95%)	No	Mean	CI (–95%)	*p*
BI_FM (kg) visit 1	82	18.32 ± 8.01	16.56–20.08	64	17.80 ± 7.30	15.97–19.62	0.820
BI_FM (kg) visit 4	55	18.29 ± 8.07	16.11–20.47	53	18.13 ± 7.18	16.15–20.11	0.907
Δ BI_FM (kg) visit 4–1	54	−0.11 ± 4.09	−1.23–1.00	51	0.01 ± 4.01	−1.11–1.14	0.890
BI_FM (%) visit 1	54	31.15 ± 4.90	29.81–32.48	51	31.23 ± 5.91	29.57–32.89	0.741
BI_FM (%) visit 4	54	29.57 ± 6.03	27.92–31.21	51	29.39 ± 6.95	27.44–31.35	0.889*
Δ BI_FM (%) visit 4–1	54	−1.58 ± 4.04	−2.68–0.47	51	−1.83 ± 4.56	−3.12–−0.55	0.951
BI_MM (kg) visit 1	82	38.46 ± 14.00	35.39–41.54	64	37.10 ± 13.65	33.69–40.51	0.586
BI_MM (kg) visit 4	56	40.12 ± 12.85	36.68–43.56	53	40.80 ± 12.92	37.24–44.36	0.886
Δ BI_MM (kg) visit 4–1	55	2.45 ±2.57	1.75–3.14	51	3.33 ± 2.75	2.56–4.10	0.091 *
BI_MM (%) visit 1	82	65.69 ±5.39	64.50–66.87	64	65.48 ± 5.03	64.22–66.73	0.350
BI_MM (%) visit 4	56	66.63 ±5.68	65.11–68.16	53	66.81 ± 6.52	65.01–68.61	0.896
Δ BI_MM (%) visit 4–1	55	1.42 ±4.02	0.33–2.51	51	1.48 ± 3.72	0.43–2.53	0.949

*p* < 0.05 Mann–Whitney U test, * t–student test for independent samples. BI—bioimpedance, FM—fat mass, Δ – delta–difference, MM—muscle mass.

**Table 4 nutrients-12-01093-t004:** Results of dual X–ray absorptiometry (DXA) measurements in studied groups.

	Vitamin D			Placebo			
	No	Mean	CI (–95%)	No	Mean	CI (–95%)	*p*
Sp BMD visit 1	83	0.76 ± 0.18	0.72–0.80	67	0.74 ± 0.18	0.70–0.79	0.447
Sp BMD visit 4	47	0.82 ± 0.19	0.76–0.87	47	0.80 ± 0.19	0.75–0.86	0.623
Δ in Sp BMD visit 4–1	47	0.04 ± 0.04	0.03–0.06	47	0.06 ± 0.04	0.05–0.08	0.025 *
TBLH BMD visit 1	83	0.87 ± 0.14	0.84–0.90	67	0.86 ± 0.15	0.82–0.89	0.672
TBLH BMD visit 4	48	0.91 ± 0.14	0.87–0.95	47	0.90 ± 0.15	0.86–0.94	0.740
Δ in TBLH BMD visit 4–1	48	0.04± 0.03	0.04–0.05	47	0.04 ± 0.03	0.03–0.05	0.504*
TFM (kg) visit 1	83	25.18 ± 9.76	23.05–27.31	67	24.29 ± 9.08	22.08–26.51	0.652
TFM (kg) visit 4	48	25.17 ± 8.72	22.63–27.70	47	25.07 ± 8.96	22.44–27.70	0.959*
Δ in TFM visit 4–1	48	0.73 ±4.55	–0.59–2.05	47	0.68 ± 4.96	–0.77–2.14	0.734
TLM (kg) visit 1	83	32.52 ± 4.55	30.04–34.99	67	31.83 ± 10.94	29.16–34.50	0.823
TLM (kg) visit 4	48	34.41 ± 10.33	31.41–37.42	47	34.48 ± 12.19	30.90–38.06	0.976 *
TFM (%) visit 1	48	43.46 ± 3.82	42.35–44.57	47	43.30 ± 4.42	42.00–44.60	0.847
TFM (%) visit 4	48	41.89 ± 5.29	40.35–43.43	47	42.32 ± 5.50	40.71–43.94	0.695
Δ in TFM (%) visit 4–1	48	–1.57 ± 4.12	–2.77–0.38	47	–0.98 ± 3.97	–2.14–0.19	0.472 *
TLM (%) visit 1	48	56.54 ± 3.82	55.43–57.65	47	56.70 ± 4.42	55.40–58.00	0.847
TLM (%) visit 4	48	58.11 ± 5.29	56.57–59.65	47	57.68 ± 5.50	56.06–59.29	0.695
Δ in TLM (%) visit 4–1	48	1.57 ± 4.12	0.38–2.77	47	0.98 ± 3.97	–0.19–2.14	0.472

*p* < 0.05 Mann–Whitney U test, * t–student test for independent samples. Sp—spine, BMD—bone mineral density, Δ—delta–difference, TBLH—total body less head, TFM—total fat mass, TLM—total lean mass.

**Table 5 nutrients-12-01093-t005:** Correlation between vitamin D levels and anthropometric variables.

	25 (OH) D Level Visit 1	25 (OH) D Level Visit 4	Δ 25 (OH) Level Visit 4–1
BMI centile 1 visit	–0.088910	**–0.232998**	–0.066400
BMI centile 4 visit	–0.128452	**–0.297140**	–0.147392
Δ in BMI centiles visit 4–1	–0.059634	–0.141315	–0.138827
BI_FM (%) visit 1	**–0.248164**	**–0.238226**	–0.020100
BI_FM (%) visit 4	**–0.213330**	**–0.226632**	–0.019981
Δ BI_FM (%) visit 4–1	0.073786	–0.013645	–0.058235
BI_MM (%) visit 1	**0.214256**	0.202164	–0.005444
BI_MM (%) visit 4	**0.204376**	**0.233062**	0.045692
Δ BI_MM (%) visit 4–1	–0.077210	0.057706	0.107305
25 (OH) D level visit 1	1.000000	**0.322476**	**–0.426874**
25 (OH) D level visit 4	**0.322476**	1.000000	**0.692743**
Δ 25 (OH) level visit 4–1	**–0.426874**	**0.692743**	1.000000
TBLH BMD visit 1	**–0.357821**	–0.164349	0.102925
TBLH BMD visit 4	**–0.377036**	–0.222562	0.012532
Δ in TBLH BMD visit 4–1	0.079692	0.007924	–0.047583
TFM (%) visit 1	–0.177243	–0.206046	–0.111781
TFM (%) visit 4	–0.035164	–0.122583	–0.069020
Δ in TFM (%) visit 2–1	0.174996	–0.008749	–0.047612
TLM (%) visit 1	0.177243	0.206046	0.111781
TLM (%) visit 4	0.035164	0.122583	0.069020
Δ in TLM (%) visit 4–1	–0.174996	0.008749	0.047612

Spearman rank correlation coefficient; bolded when *p* < 0.05, BI—bioimpedance, FM—fat mass, Δ—delta–difference, MM—muscle mass, BMD—bone mineral density, TBLH—total body less head, TFM—total fat mass, TLM—total lean mass.

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
