# Peer review of "Long-Term Effects of Vitamin D Supplementation in Obese Children During Integrated Weight–Loss Programme—A Double Blind Randomized Placebo–Controlled Trial"

_nutrients, 2020, doi:10.3390/nu12041093_

Round 1

Reviewer 1 Report

This work aims to study the impact of vitamin D supplementation during weight loss programme in overweight/obese children. The study is well conducted and clearly exposed even if it would have been nice to find the study design in supplemental datat instead to get it in Pubmed.

The study design published previously BMC Pediatrics (2017) 17:97 indicates that during this weight loss intervention programm the idea was also to control diet habits with dietetic consultations and to define with a specialist physical activity adpated to the subject.

I did not find any published results regarding these variables , how the diet habits were modified and how the pysical activity was increased (or not)

We know that vitamin D metabolism depends on the diet and especially of the lipids content in diet.

These factors desserved to be included with the results and if possible as variable of correlations.

The authors are aware of the limitations of the study and the difficulties to obtain good adhesion of obese children/adolescents to such programme and the results should  be considered as very preliminary.

Author Response

Dear Reviewer, 

thank you kindly for your review and given comments. We appreciate them a lot.

Regarding your comments:

  1. We did not add study design/protocol as it is available as an open access (it was our priority to publish as open access). Additionally it is also available in clinical trials registry https://clinicaltrials.gov/ct2/show/NCT02828228.
    Additional information regarding the background was added as requested by the Reviewer 2. (lines 77-85)
  2. Thank you for your focus on dietary habits and physical activity. In this publication we want to focus only on primary results of the study - that is on influence of vitamin D supplementation on changes in body mass (BMI), body composition and bone density. We have prepared additional publications which are now finalised and will be send to journals within next 1-2 months. We address your comments in those papers. We fully agree that this needs to be considered yet this study was primary not focused on makro/micronutrient changes in the diet. As that we believe that dietary changes were similar in both groups (placebo vs. vitamin D).
  3. We believe that for clarity of the protocol-study-result we should firstly publish primary results as planned in the study. Secondary analyses (less powerful) will be published separately. 

Thank you for your comments.

Michał Brzeziński

Reviewer 2 Report

This is an important study addressing an clear research gap: clinical trials evaluating the potential for vitamin D to support weight loss in obese children/adolescents. Unfortunately, there is a notable weakness in design that limits interpretation (point 2 below), and this feature is not adequately discussed in the manuscript (points 8 and 9 below).

1. Under Methods, more information is needed about the obesity management treatment in this manuscript, not just a comment that "Detailed information about the programme can be found in previous publications", for which no reference is provided. Specifically, the number of sessions provided, target duration of each session, clarity on whether all sessions are group vs individual, and how missed sessions were addressed (individual make-ups, or just marked as missed?).  

2. The study was powered to detect a 10% reduction in baseline BMI over 12 months occurring in 85% of intervention participants versus only 60% of control children. How was this effect size determined? Why was it powered off of BMI and not BMI percentile? For growing children/adolescents, no change in BMI over 1 year is considered a success, and will be reflected in a reduction in BMI percentile. Attaining an actual reduction in BMI is quite difficult, let alone at this magnitude (10%), and is expecting much of a vitamin. It is possible that the study was underpowered to detect smaller (more realistic) changes in BMI.

3. The authors state that 170 participants were eligible, but then that 10 were not enrolled because of vitamin D intake during the prior 3 months. That was listed as an exclusion criteria, so those participants should be considered ineligible, making the eligible denominator only 160, of which 8 declined participation. 

4. More than twice as many Vit D participants withdrew from the interventional programme than control participants. Are the reasons for this known?

5. Table 5 (correlations) is not included in the manuscript.

6. Remove the editorial instructions from the end of the Results section (This section may be divided by subheadings. It should provide a concise and precise description of the experimental results, their interpretation as well as the experimental conclusions that can be drawn.)

7. For the benefit of the reader, please start the Discussion section with a brief summary of the key findings.

8. Related to #2 above, the authors report a reduction in BMI of ~0.5 kg/m2 in the vitamin D group relative to a ~.1 kg/m2 increase in the placebo group. Similarly, BMI percentile decreased two-fold in the vitamin D group relative to placebo. These results would be bordering on clinically relevant if they were statistically significant, which suggests that this study was underpowered. This needs to be discussed.

9. Further, the authors note that prior studies (references 43-45) had mixed results on fat mass, fat-free mass, or muscle mass. Please expand this discussion, including presentation of methodology and sample sizes that may have contributed to the mixed results.

10. Discussion of prior studies examining cardiovascular factors not assessed in this study (insulin resistance, blood pressure, etc in 1st and 3rd paragraphs specifically) should be reduced and/or removed from the manuscript.

11. Given points 2, 8, and 9, the statement in the Abstract and Conclusion that this "study ads [sic] substantial results to support the thesis on no effect of vitamin D supplementation on body weight reduction in children and adolescents with vitamin D insufficiency" is a bit strong. Please revise.

Author Response

This is an important study addressing an clear research gap: clinical trials evaluating the potential for vitamin D to support weight loss in obese children/adolescents. Unfortunately, there is a notable weakness in design that limits interpretation (point 2 below), and this feature is not adequately discussed in the manuscript (points 8 and 9 below).

Response MB: Dear Reviewer thank you very much for your time and all your comments. We tried to address them best as possible. 

  1. Under Methods, more information is needed about the obesity management treatment in this manuscript, not just a comment that "Detailed information about the programme can be found in previous publications", for which no reference is provided. Specifically, the number of sessions provided, target duration of each session, clarity on whether all sessions are group vs individual, and how missed sessions were addressed (individual make-ups, or just marked as missed?).

Response MB: Additional information was added in lines 77-85 and 117-118.

  1. The study was powered to detect a 10% reduction in baseline BMI over 12 months occurring in 85% of intervention participants versus only 60% of control children. How was this effect size determined? Why was it powered off of BMI and not BMI percentile? For growing children/adolescents, no change in BMI over 1 year is considered a success, and will be reflected in a reduction in BMI percentile. Attaining an actual reduction in BMI is quite difficult, let alone at this magnitude (10%), and is expecting much of a vitamin. It is possible that the study was underpowered to detect smaller (more realistic) changes in BMI.

Response MB: The effect size was assessed on both: literature review and also our previous clinical experience with over 2000 children with overweight and obesity that we treated in “6-10-14 for Health” programme. We have seen that about 40-50% of patients had a clinical decrease on about 10% BMI centiles (z-score). We powered of on BMI as a major as inter- and intra-group results in a previously randomized group should be statistically valid. As centiles/z-scores are index/proxy measures used in children  yet we also performed analysis on BMI percentiles using Polish centile charts, as we use it clinically on daily basis. Understanding Reviewer doubts we added this information to limitation section to limitation section (319-321)

  1. The authors state that 170 participants were eligible, but then that 10 were not enrolled because of vitamin D intake during the prior 3 months. That was listed as an exclusion criteria, so those participants should be considered ineligible, making the eligible denominator only 160, of which 8 declined participation.

Response MB: We assessed patients as eligible as all those who agreed to participate in the “6-10-14 for Health” programme (-2 weeks) who fulfilled the major criteria (BMI, medical history) and had blood works performed with vit. D <30 ng/l. All of them agreed to participate in the programme. Those were 170 participants. Eight of them declined agreeing to participate in the study (0 weeks visit) and 10 previously negating vit. D supplementation were disqualified during visit (0 weeks). This is why we believe that we should report both of those groups as eligible but excluded.

  1. More than twice as many Vit D participants withdrew from the interventional programme than control participants. Are the reasons for this known?

Response MB: As the study was double-blinded we were not able to previously assess that. As we presented the participants were resigning from the whole intervention programme – not the vitamin D intervention. We were asking them at the time of resignation on the influence of the placebo/vitD. supplementation – non of the participants addressed this as a reason. More than 30-40% of participants resigns from the programme (and similar programmes) during the active phase. After unbinding the data (Nov. 2019) we once again reached out to the patients – with no additional information on the side effects of the treatment and influence on resignation from the trial.

  1. Table 5 (correlations) is not included in the manuscript.

Response MB: Please accept my apologies. I do not know how this could have happened. I have added the table 5 now – line 223.

  1. Remove the editorial instructions from the end of the Results section (This section may be divided by subheadings. It should provide a concise and precise description of the experimental results, their interpretation as well as the experimental conclusions that can be drawn.)

Response MB: Removed. thank you. (lines 233-235)

  1. For the benefit of the reader, please start the Discussion section with a brief summary of the key findings.

Response MB: Lines 240-244 were added

  1. Related to #2 above, the authors report a reduction in BMI of ~0.5 kg/m2 in the vitamin D group relative to a ~.1 kg/m2 increase in the placebo group. Similarly, BMI percentile decreased two-fold in the vitamin D group relative to placebo. These results would be bordering on clinically relevant if they were statistically significant, which suggests that this study was underpowered. This needs to be discussed.

Response MB: thank you for that comment. We were discussing that a lot in our team before the publication. Accepting and agreeing with your comment we added  lines 259-264 and 267-269  and also changed the conclusions 350-355 to soften the tone of our conclusion.

  1. Further, the authors note that prior studies (references 43-45) had mixed results on fat mass, fat-free mass, or muscle mass. Please expand this discussion, including presentation of methodology and sample sizes that may have contributed to the mixed results.

Response MB: Additional information regarding those studies was presented – lines 273-286 were added.

  1. Discussion of prior studies examining cardiovascular factors not assessed in this study (insulin resistance, blood pressure, etc in 1st and 3rd paragraphs specifically) should be reduced and/or removed from the manuscript.

Response MB: This was removed. (lines 286-287)

  1. Given points 2, 8, and 9, the statement in the Abstract and Conclusion that this "study ads [sic] substantial results to support the thesis on no effect of vitamin D supplementation on body weight reduction in children and adolescents with vitamin D insufficiency" is a bit strong. Please revise.

Response MB: The conclusions were changed as suggested – please see lines 350-355

Dear Reviewer, we hope that those changes meets your requirements. We would like to thank you for your substantial input and support. That means a lot to us.
